# Indigenous vegetables of family Cucurbitaceae of Azad Kashmir: A key emphasis on their pharmacological potential

Kulsoom Akhter[1]*, Azeem Bibi[1], Aamir Rasheed[2], Sadiq ur Rehman[1], Urooj Shafique[1], Tariq Habib[3]

**1** Department of Chemistry, The University of Azad Jammu and Kashmir, Muzaffarabad, Pakistan, **2** Department of Chemistry, University of Kotli, Kotli, Azad Jammu & Kashmir, Pakistan, **3** Department of Botany, The University of Azad Jammu and Kashmir, Muzaffarabad, Pakistan

* kalsoom.akhtar@ajku.edu.pk

## 

**Data Availability Statement:** All relevant data are within the paper.

**Funding:** The author(s) received no specific funding for this work.

## Abstract

The antioxidant capacity of extracts of different parts of Cucurbitaceae vegetables was evaluated by DPPH (2, 2-diphenyl-1-picrylhydrazyl) and ABTS (2, 2'-azino bis (ethyl benzothiazoline 6)-sulphonic acid) methods. Total phenolic content (TPC) and total flavonoid content (TFC) were also determined. The correlation of TPC, TFC, DPPH, and ABTS in different extracts of Cucurbitaceae vegetables was analyzed. The peel extracts of studied vegetables had the highest TPC, (*C. grandis* 3.00±0.86, *T. cucumerina* 3.24±0.70 and *C. moschata* 3.12±0.06 mg gallic acid equivalent (GAE) g$^{-1}$ DW) and TFC (*C. grandis* 18.96 ±1.5, *T. cucumerina* 13.92±1.41 and *C. moschata* 15.31±0.97 mg rutin equivalent (RE) g$^{-1}$ DW). The maximum antioxidant potential was obtained by the ABTS method in peel extracts of *C. grandis* (78.7%) and *C. moschata* (63.5%) while in pulp extract of *T. cucumerina* (50.1%) at 10 µg/mL. The percent radical scavenging activity (% RSA) by the DPPH method found maximum for peel and pulp of *C. grandis* (45.15 and 45.15%, respectively) and peel of *T. cucumerina* (45.15%) and *C. moschata* (34.15%). The EC50 obtained in the ABTS method was 0.54 and 7.15 µg/mL for *C. grandis* and *C. moschata*, respectively while 0.81 µg/mL for the pulp of *T. cucumerina* compared to standard ascorbic acid (1.05 µg/mL). The EC50 calculated in the DPPH method was 11.78 µg/mL, 13.34 µg/mL, and 21.00 µg/mL for *C. grandis*, *T. cucumerina*, and *C. moschata* peel respectively compared to the standard Butylated hydroxytoluene (BHT). Among each variable, the correlation between ABTS and TPC provided the highest positive correlation (r = 0.998, p< 0.05) in peel extracts.

## 1. Introduction

Intake of natural phenolic antioxidants in the diet has a positive correlation with reduced cancer mortality and heart diseases, and longer life expectancy [1–6]. These antioxidants are responsible to nurture immune function, inhibiting malignant transformation, reduce DNA

**Competing interests:** The authors have declared that no competing interests exist.

damage and lipid peroxidation [7]. Recently, growing attention has been given to screening out the natural sources of antioxidants like fruits and vegetables compared with synthetic materials [3,6,8–10]. Synthetic antioxidants like butylated hydroxytoluene (BHT) are reported to be carcinogenic and have various side effects [11].

Diseased people need to intake proper foods with functional activities for the fulfillment of their energy and nutritional needs [12]. Vegetables are the cheapest and most easily available source of nutrition which contain bioactive components including minerals, dietary fiber, and vitamins (A, C & E) as well as non-nutritive phytochemicals (carotenoids, alkaloids, terpenoids, bioactive peptides, phenolic compounds, and flavonoids) which reduce the risk of cardiovascular diseases, cancer, obesity, and diabetes [3]. Phenols, flavonoids, and vitamins were reported to contain antioxidant properties [13]. Various researchers have reported the antioxidant properties of vegetables due to the presence of phenolic content in them. However, the extent of antioxidant properties may be affected by many factors including the degree of ripeness, climatic, storage, and geographical conditions [14].

Cucurbitaceae is a large family of plants, with 130 genera and 800 species distributed in the tropical and sub-tropical regions of the world. This family is cultivated all around the world in a variety of environmental conditions having global pharmacological and dietary importance. The main producers of Cucurbitaceae are Turkey, China, India, and the USA [15]. The majority of the plants of Cucurbitaceae has medicinal importance and are used as a medicine for the remedy for ages, urinary ailments, intestinal worms, high blood pressure, kidney stone, headaches, abdominal tumors, fever, diarrhea, and skin allergies [16].

*Cucurbita moschata* is a species of the Cucurbitaceae family cultivated all around the world. It is commonly called squash or pumpkin. The pulp of *Cucurbita moschata* provides a cheaper and good source of food, having health benefits. The seeds of squash contain approximately 15.9 mg/100 g of total tocopherols, and a sufficient quantity of linoleic acid and L-tryptophan, which are largely used for the treatment of depression [17]. *Trichosanthes cucumerina* is another species widely distributed in the tropical and subtropical regions. It is commonly called snake guard, while the *Coccinia grandis* is species that grows only in the tropical region and is called ivy gourd. These two species are eaten as vegetables containing a large amount of antioxidants, anti-hypoglycemic agents, and immune systems modulators. Traditionally, these vegetables were largely used to treat leprosy, fever, asthma, bronchitis, scabies, and jaundice [18]. The main chemical constituents present in the Cucurbitaceae family are phytochemicals. The other commonly occurring compounds are carbohydrates, triterpenes, sterols, alkaloids, α-carotene, β-carotene, and lutein zeaxanthin [19].

Various methods are commonly used to measure the antioxidant activities of vegetables, including 2, 2-diphenyl-1-picrylhydrazyl radicals (DPPH) method, ferric reducing antioxidant power (FRAP) method, 2, 2′-azino-bis (3-ethylbenzthio-zoline-6)-sulphonic acid (ABTS) method, and oxygen radical absorbance capacity assay (ORAC) method [19]. Of all these methods, the most convenient and independent of expensive equipment are the DPPH and ABTS methods, thus commonly used.

The present work aimed to investigate the phenolic, flavonoid content, and antioxidant potential of various parts (peel, pulp, seeds) of some Cucurbitaceae vegetables that are commonly consumed and locally grown in Muzaffarabad, Azad Jammu & Kashmir Pakistan for their phytochemical and antioxidant behavior. Phytochemicals present in these vegetables play a protective role from oxidative stress and various other related diseases. The present study will provide information about the medicinal importance and use of studied vegetables highlighting their antioxidant properties.

## 2. Material and methods

### 2.1 Chemicals and reagents

Folinciocalteu's reagent, quercetin, gallic acid, phosphate buffer, 5, 6-Diphenyl-3-(2-pyridyl)-1, 2, 4-triazine-4′, 4″-disulfonic acid sodium salt, 2, 2'-azino-bis (3-ethylbenzothiazoline-6-sulphonic acid) (ABTS), 2, 2-diphenyl-1-picrylhydrazyl (DPPH), methanol, butylated hydroxytoluene (BHT), and potassium persulfate were purchased from Sigma-Aldrich (St. Louis, MO, USA). Ascorbic acid was obtained from Fluka (Switzerland) while sodium carbonate, sodium nitrite, aluminum chloride, and sodium hydroxide were purchased from Merck (Darmstadt, Germany). Ultrapure water was prepared using a Millipore System (Millipore, Bedford, MA, USA) and used throughout the experiments. All the used standards were stored in dark at -20°C.

### 2.2 Sample collection

Three locally grown Cucurbitaceae vegetables including *Cucurbita moschata* (Voucher No. UR-01) *Trichosanthes cucumerina* (Voucher No. UR-02) and *Coccinia grandis* (Voucher No. UR-03) were collected from the capital city Muzaffarabad of Azad Jammu and Kashmir state, Pakistan. Voucher specimens were identified by Dr. Tariq Habib, Assistant Professor, Department of Botany, University of Azad Jammu, and Kashmir (UAJ&K), Muzaffarabad, and were deposited in the herbarium of the Department of Botany, UAJ & K, Muzaffarabad.

### 2.3 Sample preparation

The collected fresh vegetables were washed with distilled water and the required parts including peel, pulp, and seeds were separated and ground into fine particles by using a mechanical grinder. To extract the antioxidant materials from the samples of vegetables, a reported method was applied with slight modification [3]. Briefly, a specific amount of sample (10.0 g) was treated with 100 mL distilled water in a shaking water bath for 30 minutes at 37°C and 100 rpm. The blended mixture was centrifuged at 4, 200 rpm for 30 minutes. The supernatant was collected, and the solvent was evaporated at room temperature to dryness. The obtained residue was weighed to calculate the extractive yield and stored in an airtight jar at -20°C for further use. The solid aqueous extracts were dissolved in distilled deionized water at a concentration of 10 mg/mL for experimental purposes.

### 2.4 Total phenolic content determination

TPC present in different parts of vegetables was determined by using the Folin-Ciocalteu method [20]. Briefly, 500 μL of the sample extract was mixed with Folin-Ciocalteu reagent (200 μL) and then mixed with 20% solution of sodium carbonate solution (1mL) after 3 minutes. The solution was incubated for one hour in the dark which turned into deep blue coloration. The absorbance of this mixture solution was monitored at 765 nm by using a blank solution prepared under the same protocol except for the extract solution. TPC was calculated from the trend line of standard gallic acid (0.05–0.25 mg/mL) and expressed in mg of GAE/g dry weight of the extract.

### 2.5 Determination of total flavonoid content

The TFC in vegetable samples was quantified according to the reported method [21]. The sample solution consisting of extract (1 mL), distilled water (2 mL), and 5% sodium nitrite (0.15 mL) was incubated at room temperature for 6 minutes. Aluminum chloride (10%, 0.15 mL) and sodium hydroxide (4%, 2 mL) were added to the sample solution followed by incubating

again for 6 minutes at room temperature. The final volume of the sample solution was made up to 10 mL by addition of ultrapure water, mixed thoroughly, and kept for 15 minutes at room temperature. Absorbance was recorded at 510 nm using UV-spectrophotometer (Shimadzu UV 1800) against the blank solution prepared under the same protocol except for the extract solution.

The total flavonoid content present in the peel, pulp, and seeds of *Cucurbitaceae* vegetables was calculated from the linear equation of quercetin (0.066–0.0166 mg/mL) taken as a standard and the results were represented as mg QE/g dry weight of the extract.

## 2.6 Antioxidant activities by ABTS method

The $ABTS^{\bullet+}$ free radical scavenging assay was used to determine the antioxidant potential of extracts [22]. The $ABTS^{\bullet+}$ free radicals in the ABTS stock solution were generated by mixing potassium persulphate (2.5 mM) and ABTS (7 mM) solution (1:1, v/v) and let to stand at room temperature for 24 hours. The solution was diluted until the absorbance (Ao) of 0.90±0.04 was obtained at 734 nm. Extracts in a range of concentrations (2.0–10.0 μg/mL) were added in $ABTS^{\bullet+}$ free radical solution and absorbance of sample solution (Ai) was measured. The percentage radical scavenging activity (% RSA) was calculated by using Eq 1, and half-maximal effective concentration (EC50 μg/mL) was calculated by trend line equation taking ascorbic acid as a standard.

$$\% \ RSA = [(Ao - Ai/Ao)] \times 100 \tag{1}$$

Where Ao is the absorbance of the blank solution, Ai is the absorbance of the sample solution.

## 2.7 Analysis of antioxidant activities by DPPH method

DPPH assay [23] was performed where different concentrations of the extracts (2–10 μg/mL) were added to 1 mL of a 0.008% methanol solution of DPPH. This solution was incubated for 30 minutes at room temperature, and UV absorbance was taken at 517 nm against a blank solution kept in the reference compartment. The control was prepared using the same protocol as described above without any extract. The BHT was used as a positive control. The percentage of DPPH free radical scavenging potential of extracts was calculated using the following equation.

$$DPPH \ radical \ scavenging \ potential \ (\%) = (1 - A_s/A_b) \times 100$$

Where $A_b$ is the absorbance of control having all reagents except the extract and $A_s$ is the absorbance of the test.

## 2.8 Statistical analysis

All experiments were performed in triplicates and the data were expressed as the mean ± standard error (SD) of three independent results. One-way analysis of variance (ANOVA) with a statistical significance level set at $p < 0.05$ correlations between the total phenolic, flavonoid content, and antioxidant capacities were made using the Pearson procedure ($p < 0.01$).

## 3. Results and discussion

### 3.1 Extraction yield

The percentage yield of aqueous extracts from the peel, pulp, and seeds of studied *Cucurbitaceae* vegetables (photographic images are included in Fig 1) were shown in Table 1. The

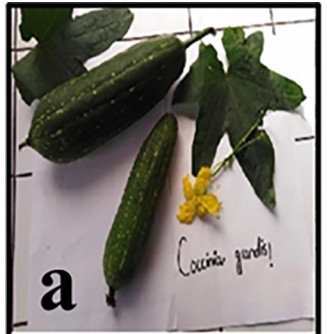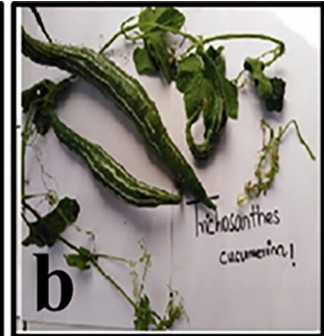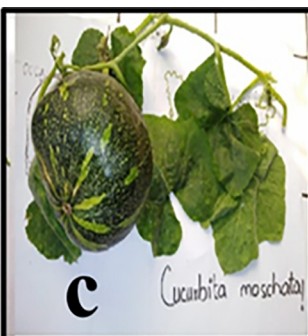

**Fig 1.** The whole fruit of a) *C. grandis*, b) *T. cucumerina* and c) *C. moschata* from the Cucurbitaceae family.

extractive yield varied among all vegetable parts where the peel extracts provided the maximum percentage yield i.e., *C. grandis* (36.80%), *T. cucumerina* (43.87%), and *C. moschata* (29.31%). Reported literature suggests that extractive yield may vary from plant to plant and depends upon the nature of secondary metabolites [24–27].

## 3.2 Evaluation of total phenolic content

The phenolic content present in plants is responsible to reduce the reactive oxygen [O] via an H atom donated by the phenolic OH group and electrons transfer from phenoxide anions [28]. The polyol enzymes that are necessary for the nutrition and health of humans are also affected by polyphenols [29]. The TPC present in different species of the Cucurbitaceae family was determined by using a standard gallic acid calibration curve and results (expressed in mg GAE/g of dry extract weight) were presented in Table 2. Khatana et al reported that the TPC in the peel of *C. moschata* extract was 6.4±0.1 mg GAE/g DW [30], which was higher than our results (3.12±0.06 mg GAE/g DW) while in the same study the total phenolic content in pulp extract was less (2.5±0.3 mg GAE/g DW) than our results (2.95±0.04 mg GAE/g DW). Kondhare and Lade reported the total phenolic composition of the aqueous extracts of *C. grandis* fruit (8.2±0.2 mg GAE/g DW) [31], which were found to be in close agreement with the sum of the phenolic content of peel, pulp, and seed extracts (8.56±2.57 mg GAE/g DW) found in the current study. Higher values of phenolic content (8.18±1.56 mg GAE/g DW) were found in *T. cucumerina* peel, pulp, and seed extracts collectively when compared to *T. cucumerina* fruit aqueous extract (4.64±0.3mg GAE/g DW) [32]. The differences in the phenolic composition of the same species reported in different studies could be due to the variation in growing

**Table 1. The percentage yield of extracts from the peel, pulp, and seeds of studied Cucurbitaceae vegetables.**

| English name of vegetable | Scientific name of vegetable | Part used | % Yield of extracts |
|---|---|---|---|
| Ivy Gourd | *Coccinia grandis* | Peel | 36.80 |
| | | Pulp | 13.70 |
| | | Seed | 6.22 |
| Snake Gourd | *Trichosanthes cucumerina* | Peel | 43.87 |
| | | Pulp | 22.43 |
| | | Seed | 25.60 |
| Butternut squash | *Cucurbita moschata* | Peel | 29.31 |
| | | Pulp | 25.56 |
| | | Seed | 12.42 |

**Table 2. TPC and TFC in the aqueous extracts of various parts of studied *Cucurbitaceae* vegetables.**

| Vegetable Scientific name | TPC (mg GAE/g dry extract) | | | TFC (mg RE/g dry extract) | | |
|---|---|---|---|---|---|---|
| | **Peel** | **Pulp** | **Seed** | **Peel** | **Pulp** | **Seed** |
| *Coccinia grandis* | 3.00±0.86 | 2.87±0.91 | 2.69±0.83 | 18.96±1.5 | 9.50±0.90 | 7.76±0.20 |
| *Trichosanthes cucumerina* | 3.24±0.70 | 2.51±0.41 | 2.43±0.45 | 13.92±1.41 | 9.14±0.46 | 8.73±0.05 |
| *Cucurbita moschata* | 3.12±0.06 | 2.95±0.04 | 2.88±1.6 | 15.31±0.97 | 14.38±0.73 | 9.71±0.16 |

conditions, indicating that phenolic content may vary with variation in geographical region and climatic conditions.

### 3.3 Evaluation of total flavonoid content

Flavonoids are among the polyphenols, which gained significant interest because of their various biological effects as anti-inflammatory, antimicrobial, anticarcinogenic, antioxidant, and vasorelaxant [33]. Flavonoids have a broad spectrum of antioxidant properties particularly due to radical scavenging activity.

The highest flavonoid content was found in the peel extracts of all studied *Cucurbitaceae* vegetables (Table 2) where *C. grandis*, *T. cucumerina*, and *C. moschata* exhibited 18.96±1.5, 13.92±1.4, and 15.31±0.97 mg RE/g DW flavonoid content respectively. The lowest flavonoid content was obtained from the seed extracts., *C. grandis* (7.76±0.20 mg RE/g DW), *T. cucumerina* (8.73±0.05 mg RE/g DW), and *C. moschata* (9.71±0.16 mg RE/g DW). No reported literature was available for the total flavonoid content of any species of the *Cucurbitaceae* family.

### 3.4 Antioxidant activities by ABTS method

The ABTS method has been broadly used for the assessment of the antioxidant capabilities of natural products. In this method, the dark blue color of 2, 2′-azino-bis (3-ethylbenzothiazoline-6-sulfonate) radical cation (ABTS$^{\bullet+}$), is reduced by an antioxidant (which transfer an electron) into colorless ABTS solution. The change in concentration of ABTS$^{\bullet+}$ after the reduction process is measured spectrophotometrically. In the current study, the antioxidation potential of the peel, pulp, and seed extracts from three *Cucurbitaceae* vegetables was evaluated and presented in Table 3. The EC50 (concentration of extract scavenging 50% of free radicals) was calculated from the % RSA values by using the linear equation of ascorbic acid taken as a standard (Table 5).

**Table 3. Antioxidant activity of aqueous extracts of various parts of studied *Cucurbitaceae* vegetables evaluated by ABTS•+ radical scavenging method.**

| Conc e- ntrat ion μg/mL | % ABTS Radical Scavenging Activity | | | | | | | | | |
|---|---|---|---|---|---|---|---|---|---|---|
| | Ascor bic acid | *Coccinia grandis* | | | *Trichosanthes cucumerina* | | | *Cucurbita moschata* | | |
| | | **Peel** | **Pulp** | **Seed** | **Peel** | **Pulp** | **Seed** | **Peel** | **Pulp** | **Seed** |
| 2.0 | 53.0±1.05 | 53.2±0.92 | 51.4±0.99 | 25.5±0.32 | 27.4±0.21 | 27.6±0.43 | 14.7±0.09 | 27.1±0.89 | 26.9±0.85 | 4.7±0.03 |
| 4.0 | 68.0±1.08 | 60.9±0.75 | 60.8±0.82 | 27.5±0.40 | 31.0±0.25 | 35.1±0.50 | 15.2±0.02 | 37.9±0.98 | 29.7±0.92 | 10.8±0.32 |
| 6.0 | 80.0±1.20 | 69.9±0.93 | 64.9±0.87 | 28.3±0.69 | 34.2±0.45 | 41.0±0.67 | 18.1±0.53 | 46.7±1.06 | 32.6±0.97 | 17.8±0.50 |
| 8.0 | 92.0±1.21 | 72.1±1.15 | 69.9±0.92 | 30.2±0.72 | 38.7±0.70 | 47.4±0.58 | 19.3±0.45 | 50.3±1.21 | 38.8±1.5 | 20.9±0.65 |
| 10.0 | 98.0±1.30 | 78.7±1.05 | 74.4±1.20 | 32.1±0.77 | 40.9±0.68 | 50.1±0.98 | 20.9±0.91 | 63.5±1.34 | 42.2±1.30 | 25.9±0.89 |

**Table 4. The antioxidant activity of the aqueous extracts of various parts of studied *Cucurbitaceae* vegetables evaluated by the DPPH radical scavenging assay.**

| Concentration µg/mL | % DPPH radical scavenging assay | | | | | | | | | |
|---|---|---|---|---|---|---|---|---|---|---|
| | Butylated hydroxy toluene (BHT) | *Coccinia grandis* | | | *Trichosanthes cucumerina* | | | *Cucurbita moschata* | | |
| | | Peel | Pulp | Seed | Peel | Pulp | Seed | Peel | Pulp | Seed |
| 2.0 | 53.0±1.05 | 22.46±0.21 | 21.61±0.84 | 18.69±0.47 | 28.53±0.34 | 18.53±0.98 | 15.3±0.95 | 22.76±0.51 | 18.3±0.22 | 6.23±0.06 |
| 4.0 | 68.0±1.08 | 29.76±0.61 | 27.38±0.95 | 24.69±0.90 | 30.69±0.55 | 26.46±0.87 | 20.84±0.65 | 25.23±0.59 | 21.5±0.81 | 13.07±0.05 |
| 6.0 | 80.0±1.20 | 33.46±0.70 | 32.3±1.02 | 29.76±0.85 | 32.15±1.01 | 32.07±1.50 | 26.15±0.38 | 28.04±0.98 | 24.76±0.45 | 16.3±0.62 |
| 8.0 | 92.0±1.21 | 39.53±1.09 | 37.0±1.15 | 34.46±0.55 | 38.38±0.65 | 35.15±1.00 | 29.76±0.78 | 31.26±1.02 | 27.76±0.90 | 19.01±0.59 |
| 10.0 | 98.0±1.30 | 45.15±1.10 | 45.15±1.08 | 40.84±1.09 | 45.15±1.18 | 37.46±1.20 | 32.15±1.15 | 34.22±1.05 | 30.65±1.11 | 23.05±0.43 |

The highest antioxidant property was presented by the peel extracts of three Cucurbitaceae vegetables. *C. grandis* and *C. moschata* exhibited the scavenging activity in the range of 53.2±0.92–78.7±1.05% and 27.1±0.89–63.5±1.34% respectively in the concentration range of 2–10 µg/mL. Whereas *T. Cucumber* in a pulp extract provided the radical scavenging activity in the range of 27.6±0.43–50.1±0.98% with the same concentrations. A moderate level of % scavenging radical activities was noted in the pulp extracts of *C. grandis* (51.4±0.99–74.4±1.20%), *C. moschata* (26.9±0.85–42.2±1.30%), and peel extracts of *T. cucumerina* (27.4±0.21–40.9±0.68%). The lowest scavenging potential was exhibited by the seed extracts of three studied samples. A similar trend was reported by Xu et al. [34] and Zhang et al. [35] where peel extract provided the maximum antioxidant potential than pulp and seeds. Variation in antioxidant potential among different fruit tissues of the same vegetable could be due to the presence of the maximum amount of phenolic and flavonoid content in the peel.

A group of researchers reported the % ABTS radical scavenging potential of *C. grandis* whole fruit extracts prepared in different solvents including petroleum ether (43.2±0.2), dichloromethane (61.0±0.5), acetone (57.4±0.4), methanol (95.8±1.0), and water (88.4±0.6) at a concentration of 100 µg/mL [31]. In the present study, almost the same results were observed by 10 µg/mL of peel and pulp extracts indicating that the *C. grandis* species in the current work has far greater antioxidant potential. This can be due to the various number of antioxidants present in the same species grown in different areas under different conditions. No study was found in the reported literature regarding the % RSA of *T. cucumerina* and *C. moschata* measured by the ABTS method.

**Table 5. Means EC50 values for ABTS•+ and DPPH radical scavenging potential of the aqueous extracts of various parts of studied *Cucurbitaceae* vegetables.**

| Sample | Vegetable Part | EC50 for ABTS•+radical scavenging potential (µg/mL) | EC50 for DPPH radical scavenging potential (µg/mL) |
|---|---|---|---|
| Ascorbic acid | - | 1.05 | - |
| BHT | - | - | 7.35 |
| *Coccinia grandis* | Peel | 0.54 | 11.78 |
| | Pulp | 0.81 | 12.10 |
| | Seed | 32.76 | 13.51 |
| *Trichosanthes cucumerina* | Peel | 14.96 | 13.34 |
| | Pulp | 9.40 | 14.62 |
| | Seed | 45.22 | 17.80 |
| *Cucurbita moschata* | Peel | 7.15 | 21.0 |
| | Pulp | 14.04 | 22.41 |
| | Seed | 18.09 | 23.41 |

**Table 6. Pearson correlation among EC50 of DPPH and ABTS radical scavenging activity and TPCs and TFCs.**

| Correlation (r) among variable | Sample | TPC | TFC |
|---|---|---|---|
| **ABTS** | Peel | 0.998** (0.030) | -0.954 ns (0.127) |
| | Pulp | -0.92** (0.02) | -0.915** (0.058) |
| | Seed | 0.260 ns (0.23) | 0.984 ns (0.876) |
| **DPPH** | Peel | 0.234** (0.012) | -0.47 ns (0.93) |
| | Pulp | 0.24** (0.005) | 0.87 ns (0.09) |
| | Seed | 0.66** (0.01) | 0.959 ns (0.07) |

r, correlation coefficient; TPC, total phenolic content; TFC, total flavonoid content; DPPH, DPPH radical scavenging activity; ABTS, 2, 2´-azino-bis (3-ethylbenzthiozoline-6)-sulphonic acid method. The numbers in parentheses are p-values.

**Correlation is significant at the 0.01/0.05 level (two-tailed), ns = not significant.

## 3.5 Antioxidant activities by DPPH method

The antioxidant ability of any material can be evaluated by various methods, in which the DPPH essay is very simple and rapid. This method is frequently used to assess the ability of a material to scavenge the free radical and provides valuable information about hydrogen donors compounds [36]. Free electron generated by DPPH reagent show absorption peak at 517 nm [37]. However, when the odd electron of the DPPH reagent is reduced by the H atom of any hydrogen donor substance, the purple color of DPPH fades to yellow. The color change is the indication of the generation of reduced DPPH [38]. The extent of decolorization of DPPH indicates the radical-scavenging potential of the antioxidant. The resulting decoloriza-tion is stoichiometric with respect to the number of electrons captured. Results of percent scavenging activity and the obtained EC50 values were presented in Tables 4 and 5 respectively.

DPPH radical scavenging ability increased as the concentration of extracts was increased. The percent DPPH radical scavenging potential (% RSA) of peel extracts of all studied vegetables was found to be maximum which might be attributed to the greater amount of phenols and flavonoids present in peel extracts. The % RSA of the aqueous peel extracts ranged for *C. grandis* (22.46±0.21–45.15±1.10%), the pulp (21.61±0.84–45.15±1.08%) and seed (18.69±0.47–40.84 ±1.09%), *T. cucumerina* peel (28.52±0.34–45.15±1.18%), pulp (18.53±0.98–37.46±1.20%), and seed (15.3±0.95–32.15±1.15%), *C. moschata* peel (22.76±0.51–34.22±1.05%), pulp (18.3±0.22–30.65±1.11%) and seed (6.23±0.06–23.05±0.43%) at the concentration range of 2.0–10.0 μg/mL. The order for the DPPH radical scavenging activity of the tissues of all studied vegetables was peel > pulp > seed. Liyanage et al reported the DPPH radical scavenging potential of the aqueous extracts of *T. Cucumerina* fruit, leaves, and flower with the % RSA values of 10.83 ± 0.7, 3.08 ± 0.2, and 4.16 ± 0.1 respectively, at a concentration of 100 μg/mL [32].

## 3.6 Statistical analysis

To identify the chemical content responsible to give the antioxidant capacity to the vegetables of the Cucurbitaceae family, correlation coefficients among the TPC, TFC, and antioxidant capacity by ABTS and DPPH methods were analyzed. Table 6 compiled the Pearson

correlation as indicated with the coefficient of correlation (r) values among IC50. The higher the r-value, the higher the correlation of variables. In the case of peel extract, among each variable, the correlation between ABTS and total phenolic content showed the highest positive correlation (r = 0.998, p<0.05) which was like the study of other researchers [39] for the cucurbit family while in the study of seed the ABTS and TPC had statistically no significant correlation. In the case of the seed sample, a moderate and positive correlation between DPPH radicle scavenging capacity and TPC (r = 0.686, p<0.05) was observed. In another report, the highest and most positive correlation exists between DPPH scavenging capacity and total flavonoid (r = 0.910, p<0.01) for the species of the Cucurbitaceae family [40].

For peel, ABTS values were correlated with the phenolic content (r = 0.998). The TFC and ABTS in the case of peel were not significantly correlated as p values were higher than the 0.05 significance level. However, studies have been found to report a high correlation between ABTS values with the total phenolic (TP) and total flavonoid (TF) content of other Cucurbitaceae species [39]. The difference may arise due to the difference in specific solvents used. Other compounds could be involved in the antioxidant capacity apart from phenols, such as vitamin C. For pulp the ABTS have negative correlation with TPC and TFC (r = -0.92 and -0.915 respectively). In the case of seed, ABTS was not significantly correlated with the phenolic content and flavonoid content (p> 0.05). For peel, pulp, and seed, DPPH values were correlated with the phenolic content, while TFC was not significantly correlated with the EC50 value of DPPH (p> 0.05). There is no correlation between DPPH and TFC might be due to different mechanism abilities and the use of different standards. While the contradictory report also found such as the antioxidant activities of *T. cucumerina* well correlated with the amount of total phenolic and flavonoid contents [32]. This study indicated that phenolic compounds significantly affected all antioxidant activities while flavonoids did not contribute significantly.

## 4. Conclusion

To evaluate the antioxidant capacity of the sample, various methods must be used in parallel. The aqueous extracts of peel, pulp, and seed had the lowest EC50 values in the ABTS method and were considered strong antioxidants. The correlation between ABTS and TPC exhibited the highest positive correlation in peel extracts. Phenolic compounds were the major contributors to ABTS antioxidant capacity in peel extracts. Not all variables in the peel, pulp, and seed extracts from three species of Cucurbitaceae were linear. Results showed that the vegetables of the family *Cucurbitaceae* have a high potential for use in pharmacy and phytotherapy. It could be concluded that the studied vegetables are natural sources of phytochemicals and antioxidant substances of high biological importance. Antioxidant activities of the extracts may be attributed to these phytochemicals. The edible part of the vegetable is its pulp, but our results indicated the presence of maximum phenolic and flavonoid content in the peel of all studied vegetables based on which, we strongly recommend the use of the peel of all studied vegetables as an economical and valuable source of phytochemicals and antioxidants rather than discarded as waste material. Moreover, in view of phytochemical screening and antioxidant potential presented by these vegetables of the Cucurbitaceae family, their optimum consumption in the diet has been strongly recommended to reduce the risk of oxidative stress and its related diseases.

## Acknowledgments

The authors are thankful to the University of Azad Jammu and Kashmir, Muzaffarabad for providing a lab facility to carry out this research work.

## Ethics statement

This research did not include any human subjects or animal experiments.

## Author Contributions

**Conceptualization:** Kulsoom Akhter.

**Formal analysis:** Kulsoom Akhter, Azeem Bibi, Urooj Shafique.

**Investigation:** Kulsoom Akhter, Aamir Rasheed, Sadiq ur Rehman, Tariq Habib.

**Methodology:** Kulsoom Akhter, Urooj Shafique.

**Supervision:** Kulsoom Akhter.

**Visualization:** Kulsoom Akhter.

**Writing – original draft:** Kulsoom Akhter, Azeem Bibi, Urooj Shafique.

**Writing – review & editing:** Kulsoom Akhter, Azeem Bibi, Urooj Shafique.

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
