## [Decision Letter · Decision Letter 0]

14 Feb 2022

PONE-D-21-38940Indigenous vegetables of family Cucurbitaceae of Azad Kashmir: A key emphasis to their pharmacological potentialPLOS ONE

Dear Dr. ,

Thank you for submitting your manuscript to PLOS ONE. After careful consideration, we feel that it has merit but does not fully meet PLOS ONE’s publication criteria as it currently stands. Therefore, we invite you to submit a revised version of the manuscript that addresses the points raised during the review process.

We look forward to receiving your revised manuscript.

Kind regards,

Muhammad Ishtiaq

Academic Editor

PLOS ONE

Journal Requirements:

Whilst you may use any professional scientific editing service of your choice, PLOS has partnered with both American Journal Experts (AJE) and Editage to provide discounted services to PLOS authors. Both organizations have experience helping authors meet PLOS guidelines and can provide language editing, translation, manuscript formatting, and figure formatting to ensure your manuscript meets our submission guidelines. To take advantage of our partnership with AJE, visit the AJE website (http://aje.com/go/plos) for a 15% discount off AJE services. To take advantage of our partnership with Editage, visit the Editage website (www.editage.com) and enter referral code PLOSEDIT for a 15% discount off Editage services.  If the PLOS editorial team finds any language issues in text that either AJE or Editage has edited, the service provider will re-edit the text for free.

“I have read the journal's policy and the authors of this manuscript have no competing interests”

5. PLOS requires an ORCID iD for the corresponding author in Editorial Manager on papers submitted after December 6th, 2016. Please ensure that you have an ORCID iD and that it is validated in Editorial Manager. To do this, go to ‘Update my Information’ (in the upper left-hand corner of the main menu), and click on the Fetch/Validate link next to the ORCID field. This will take you to the ORCID site and allow you to create a new iD or authenticate a pre-existing iD in Editorial Manager. Please see the following video for instructions on linking an ORCID iD to your Editorial Manager account: https://www.youtube.com/watch?v=_xcclfuvtxQ.

Reviewers' comments:

Reviewer's Responses to Questions

**Comments to the Author**

1. Is the manuscript technically sound, and do the data support the conclusions?

Reviewer #1: Yes

Reviewer #2: Partly

Reviewer #3: No

2. Has the statistical analysis been performed appropriately and rigorously? 

Reviewer #1: Yes

Reviewer #2: Yes

Reviewer #3: No

3. Have the authors made all data underlying the findings in their manuscript fully available?

Reviewer #1: Yes

Reviewer #2: Yes

Reviewer #3: Yes

4. Is the manuscript presented in an intelligible fashion and written in standard English?

Reviewer #1: Yes

Reviewer #2: Yes

Reviewer #3: No

5. Review Comments to the Author

Reviewer #1: 1.Why selected these plants from the whole family?

2. Provide the importance and recommendations of the study.

3. Cross check the reference with main body text.

4. Typographical and English language corrections need improvement.

Reviewer #2: The abstract is very concise but abbreviation used more. So, correct it. Introduction is very brief. it does not cover whole topic. Add relevant data in introduction in sequence. Methods are designed in good form. Results description showed lack of justifications. Please add suitable discussion. Conclusion is not properly indicated key findings. All References must be formatted according to given format of the journal. Some spelling mistakes are observed. So, I think, major revision of the paper is required

Reviewer #3: Indigenous vegetables of family Cucurbitaceae of Azad Kashmir: A key emphasis to their pharmacological potential

I congratulate authors for presenting a nice MS. The MS is well written. However there are some issues to be corrected before publishing.

Grammatical mistake

µgm/ mL;

Genus and species must have gap;

Mistake in ref 15 and 26

Please make reference crosschecking for similar type of work,

What is the difference between this work and previous works (Ref 24, Ref 25, Ref 26, Ref 31)?

What is the reason for positive, negative and no correlation of peel, pulp and seed. Is there any evidence from this research?

What is take home message from this study? Please state this nutshell in conclusion.

What is the global/regional inferences of this study?

All the best!

6. PLOS authors have the option to publish the peer review history of their article (what does this mean?). If published, this will include your full peer review and any attached files.

Reviewer #1: No

Reviewer #2: No

Reviewer #3: No

---

## [Author Response · Author response to Decision Letter 0]

14 Mar 2022

yes. we have accommodate all possible changes

---

## [Editor Report · Decision Letter 1]

11 Apr 2022

PONE-D-21-38940R1Indigenous vegetables of family Cucurbitaceae of Azad Kashmir: A key emphasis to their pharmacological potentialPLOS ONE

Dear Dr. SB,

Thank you for submitting your manuscript to PLOS ONE. After careful consideration, we feel that it has merit but does not fully meet PLOS ONE’s publication criteria as it currently stands. Therefore, we invite you to submit a revised version of the manuscript that addresses the points raised during the review process.

ACADEMIC EDITOR: Please insert comments here and delete this placeholder text when finished. Be sure to:

Please correct paper as guided now for improvement, see comments section

We look forward to receiving your revised manuscript.

Kind regards,

Muhammad Ishtiaq

Academic Editor

PLOS ONE

Journal Requirements:

Additional Editor Comments (if provided):

The paper albeit has been updated as per reviewers' comments but it needs further improvements. Please correct as:

1) -- Table 1, see percentage yield is not correct, it should be revised

2) ---Table 2, write full name of plant in Latin

3)- -Table 3, write full name of plant in Latin

4) revise English language, get expertise from Native English speaker or from any Professor of English Deptt of your University and also add his letter
---

## [Author Response · Author response to Decision Letter 1]

25 Apr 2022

Thank you for the opportunity to submit a revised version of the manuscript. As requested, we provide here a rebuttal to the comments, as well as a detailed description of how we have met the suggestions for improvement. Please note that we have made almost the proposed changes. 

Thanks again for your consideration.

---

## [Editor Report · Decision Letter 2]

23 May 2022

Indigenous vegetables of family Cucurbitaceae of Azad Kashmir: A key emphasis on their pharmacological potential

PONE-D-21-38940R2

Dear Dr. Kalsoom Akhter ,

We’re pleased to inform you that your manuscript has been judged scientifically suitable for publication and will be formally accepted for publication once it meets all outstanding technical requirements.

Kind regards,

Muhammad Ishtiaq

Academic Editor

PLOS ONE

Additional Editor Comments (optional):

The paper is well written and properly updated
---

## [Editor Report · Acceptance letter]

27 May 2022

PONE-D-21-38940R2 

Indigenous vegetables of family Cucurbitaceae of Azad Kashmir: A key emphasis on their pharmacological potential 

Dear Dr. Akhter:

I'm pleased to inform you that your manuscript has been deemed suitable for publication in PLOS ONE. Congratulations! Your manuscript is now with our production department. 

Kind regards, 

on behalf of

Dr. Muhammad Ishtiaq 

Academic Editor

PLOS ONE